# The Prime Beat Components Extraction Method for the Time Spectra Analysis of Nuclear Resonant Forward Scattering

**DOI:** 10.3390/ma12101657

**Published:** 2019-05-21

**Authors:** Tang Li, Xiaowei Zhang

**Affiliations:** 1Beijing Synchrotron Radiation Facility, Institute of High Energy Physics, Chinese Academy of Sciences, Beijing 100049, China; litang@ihep.ac.cn; 2School of Physical Sciences, University of Chinese Academy of Sciences, Beijing 100049, China

**Keywords:** nuclear resonant forward scattering, quantum beats, Mössbauer spectroscopy, prime beat components, hyperfine structure, synchrotron radiation

## Abstract

Spectra of quantum beats (QBs) of nuclear resonant forward scattering contain the interference information of all allowed energy transitions of a nucleus, which makes it complicated to extract hyperfine structure directly. Here, we propose a new method, based upon the extraction of prime beat components, to understand QBs. In this method, the origin of major spectral lines in the Fourier Transformation of QBs is studied, and the energy levels of hyperfine structure are obtained directly from the QBs. We applied this method to the temperature dependent QBs of hematite. The Morin temperature and hyperfine structure obtained by this method are in consistent with that by conventional Mössbauer spectroscopy (MS). It serves to interpret the components of QBs of nuclear resonant forward scattering as simple as the conventional (MS).

## 1. Introduction

Mössbauer discovered the recoilless γ-ray resonant absorption of ^191^Ir nuclei in 1958 [1]. The technique of Mössbauer spectroscopy opens the door to the exploration of the local electronic and magnetic structure via hyperfine interactions [2]. There are two main branches of Mössbauer spectroscopy. One is the energy-domain Mössbauer spectrum (MS). The γ-ray emission/absorption spectrum of resonant nuclei is measured in a sample containing resonant nuclei by the Doppler oscillation of a γ-ray source. The γ-ray source has two options: conventional radioactive isotope or synchrotron radiation [3,4]. The energy-domain Mössbauer spectrum illuminates the nuclear hyperfine structure directly from the resonance line absorption/emission positions and is relatively easy to analyze. The other branch is the time-domain MS which reflects the hyperfine structure (HFS) due to the coherent process of nuclear resonant forward scattering (NFS). The time-domain MS, also called the spectrum of quantum beats of nuclear forward scattering (NFS), is mostly carried on synchrotron radiation (SR). In 1986, Gerdau et al. introduced the nuclear resonant spectroscopy in the time-domain [5]. Since then, this experimental technique has been intensively developed [6,7,8]. Because synchrotron radiation has the advantages of well-collimation, high brilliance, pulsed structure and ideally polarization, it has broadened the Mössbauer nuclei without γ-ray isotope, and some research is carried in both the energy and time domain [8,9,10,11,12].

Here, we focus on the time-domain MS data analysis method. The time spectrum of NFS was first considered theoretically by Kagan et al. [13]. Trammel and Hannon explained the multiple nuclear resonances—quantum beats for the first time [14,15]. Because NFS spectrum reflects the interference between narrow energy levels and is superposed on a Bessel function modified exponential decay curve [16], the comprehension of the NFS spectrum is not as easy as the energy-domain MS. One way to analyze the spectrum is to calculate the reflected or transmitted amplitudes as a function of energy and perform Fourier transformation from the frequency domain to time domain and fit with experimental data. The representative software is CONUSS (Version 2.2.0, Wolfgang Struhahn, Pasadena, CA, USA) [17]. Another way is to calculate the time-domain MS directly by the nuclear resonance scattering theory. The representative software is MOTIF [18]. These methods focus on fitting or calculation of the time spectra. They are sophisticated in analyzing the NFS data, requiring prior experiences for the initial guess and high skills.

Aiming at a quick analysis of NFS data as simple as conventional Mossbauer spectroscopy, we propose a method by direct conversion of the time-domain spectrum into energy lines spectrum and interpreting the interference information of different sublevels’ energy transition. To the best of our knowledge, there is still no relevant research work on interpreting the origins of these spectral lines. Different from the method of transforming QBs to a frequency spectrum with an adaptable window [19], the treatment of data is to convert the QBs to its frequency space by the maximum entropy method (MEM) [20]. This frequency information is explained by a method named “prime beat components (PBCs) extraction method”, which is inspired by the spectrum analysis of time series in signal processing [21] and based on the principle that the NFS is a coherent process. We introduce the consideration of the PBC extraction method. By this method, the PBCs are extracted from the energy lines spectra that convert from QBs. Most of the spectral lines with higher intensity, which are linear combinations of PBCs, are assigned after conversion and calculation. The PBC extraction method is applied to the NFS spectra of a hematite powder sample. We extract the prime beat components and assign the beat components in the energy lines spectra. Most of the primary spectral lines are assigned by linear combinations of four PBCs. Then, we analyze the NFS spectra of hematite at various temperatures. The differences between energy levels are in accordance with the extracted prime beat components, the value of which in excited energy levels reverse below and above the Morin temperature. The results are in good agreement with conventional MS experiments.

## 2. Methods and Materials

### 2.1. Consideration of the Time-Domain NFS Spectra Analysis

There are two main problems in the analysis of the NFS time spectra. One problem is the lack of the phase information of scattered photon field in measured time spectra. This problem can be solved by using the LLL-interferometer and a reference sample to solve [22]. The other problem is to resolve the mixing of quantum beats, which is caused by hyperfine interactions, and dynamical beats, which are caused by the effective thickness. The dynamical beats can be understood as the intra-resonance interference between different spectral components with the resonance line. By contrast, quantum beats can be understood as the inter-resonance interference between different resonances in the nucleus. The hybrid beats have been studied in detail in [23]. The dynamical beats depending on effective thickness can be described by exponential or Bessel function. In this paper, we mainly consider the quantum beats of NFS spectra after eliminating the influence of dynamic beats.

Here, ^57^Fe nucleus is taken as an example, and the six transition lines between different energy levels are shown in Figure 1a. In Figure 1b, the conventional MS of Fe_2_O_3_ is drawn and it is convenient to extract the Mössbauer nucleus transition levels from this spectrum. However, the time-domain NFS spectrum contains the interference fringes between energy transition levels of a nucleus and is pretty hard to understand intuitively. There are two ways to analyze the time spectra. One is to treat this problem as a forward problem. The idea of these methods is to fit the NFS spectra. The representative software is CONUSS, MOTIF and so on. These methods need to deal with the initial guess of parameters during fitting. The other way is to treat this problem as an inverse problem. The idea of this method is to deal with the interference among different energy transitions directly after converting the time spectra to energy space by Fourier transform. However, the difficulty is to analyze the interference fringes among different nuclear transition levels.

In this paper, we propose a method to understand the meaning of these interference fringes and find a reasonable solution that is related to the HFS of nucleus. The interference information is the linear combinations among the energy transitions. The solution can be called “Prime Beat Component” (PBC). Mathematically, the PBCs are defined as the shortest list of beat components that can represent the origin of the most spectral lines in the converted spectrum. Physically, the PBCs mean the intervals between the energy levels of the hyperfine structure. The number of PBC is determined by the hyperfine structure (HFS) of the nucleus. For ^57^Fe nucleus, there are six allowed transitions as shown in Figure 1a. In that model, the PBCs of ^57^Fe nucleus can be described by a, b, c, d in Figure 1a according to the definition above. For an unknown sample, the number of PBCs increase sequentially until the shortest list of PBCs can explain most of the spectral lines with minimum average deviations. Then, the interference components in the energy-domain are analyzed as the conventional MS.

### 2.2. Obtaining Interference Fringes and Beat Components

Before converting the spectrum to its frequency space, it is necessary to filter the baseline and keep the interference information to ensure the accuracy after time-frequency conversion. This procedure is like the data analysis in Extended X-ray Absorption Fine Structure [25].

The purpose of this step is to obtain the interference information independently. Therefore, we need to renormalize the quantum beats by dividing the baseline. The baseline of the NFS spectra can be fitted by exponential function or Bessel function after the logarithm. According to the principle that a polynomial can expand the exponential and Bessel function, we fit the baseline of quantum beats by polynomials. Then, the interference fringes can be obtained by the residual between the logarithm of NFS spectra and the fitting baseline. This operation is equal to renormalizing the original NFS spectra. Thus, the information of quantum beats can be kept.

After the interference fringes are obtained, they are transformed from the time domain to the frequency space where the interference information converts to the beat components. In spectral analysis, the Fourier transform is the standard method implemented in time-frequency conversion. Instead of Fourier transform, we use the Maximum Entropy Method (MEM) here. One advantage of MEM is a better resolution than Fourier Transform. The philosophy of entropy maximization dictates that it makes no assumptions about missing data. The MEM fills in gaps or extends the data in a way which is consistent with the available data but also in such a way that entropy is maximized. Therefore, the MEM is more convenient and accurate than Fourier Transform [26]. There are three categories of the MEM method: MEM1, MEM2, and GMEM. In our cases, we utilize the MEM1 method which is mostly applied on spectral analysis. In this paper, the 1D MEM1 commercial software called MemCalc Version 1.2 (GMS Co., Tokyo, Japan) is implemented to transfer the quantum beats to the frequency space and obtain the beat components of the QBs.

After we obtained the frequency spectrum, we convert the frequency to the energy unit using a factor of 1 Hz = 4.1356 × 10^−15^ eV. In these spectral lines, there are beat components in energy domain which correspond to the interference fringes in the QB spectrum.

### 2.3. Meaning of QB Components and the Prime Beat Components’ Extraction

Though the spectral lines in the energy domain are obtained, it is vital to find the PBCs and pick out the beat components in those spectral lines which can be interpreted by PBCs. In addition, the PBCs can reflect the hyperfine structure of the nucleus interference. Therefore, in this section, we need to narrow down the range of spectral lines and extract the PBCs to depict the beat components of linear combinations by PBCs with minimum residue.

We assume that all components of the interference among different energy transition levels can be found in these spectral lines. It is believed that there is a list of PBCs that could explain the other spectral lines with their linear combinations. Here, an exhaust algorithm is proposed to pick up the PBCs among hundreds of spectral lines for assigning the rest of the beat components and determining the HFS.

The first step is to pick out the spectral lines that can be assigned as beat components. The number of beat components depends on the number of PBCs or the number of energy gaps in HFS. For example, if there are 2 PBCs in a nucleus HFS model as shown in Figure 2, it means that there are four energy transitions ω_1_, ω_2_, ω_3_, ω_4_, and two energy gaps a = (ω_3_ – ω_1_) or (ω_4_ – ω_2_), b = (ω_2_ – ω_1_) or (ω_4_ – ω_3_). As the number of PBCs increases, if the number of beat components equals *n*, the number of all possible combinations by PBCs is $(3^{n}-1)/2\;$. It can be depicted by Formula (1). In Formula (1), the first matrix describes the coefficients of PBCs in the linear combinations and the coefficients can be chosen from the numbers 0, −1, and 1. In addition, its matrix size is $\frac{3^{n}-1}{2}\times n$. The second vector describes the PBC as a, b, c, d, …, and the size is *n*.
(1)$$\begin{array}{cc} (3^{n}-1)/2\;\;\mathrm{combinations}\;\mathrm{by}\;n\;\mathrm{prime}\;\;\mathrm{beat}\;\mathrm{components} & \;\{{\left(\begin{array}{cccc} 1 & 0 & \cdots & 0\\ 0 & 1 & \cdots & 0\\ \vdots & \vdots & \cdots & \vdots \\ 1 & 1 & \cdots & -1\\ \vdots & \vdots & \vdots & \vdots \\ 1 & 1 & \cdots & 1 \end{array}\right)}_{\frac{3^{n}-1}{2}\times n}{\left(\begin{array}{c} a\\ b\\ c\\ \vdots \end{array}\right)}_{n\times 1} \end{array}$$

Taking ^57^Fe nucleus HFS as an example, the number of the PBCs is four, and we use a, b, c, d for representation in Figure 1. Hence, there are (3^4^ − 1)/2 = 40 linear equations that are used as constraints to restrain the value of PBCs and the six energy transition lines of ^57^Fe nucleus are described as b, b + c, a + b, a + b + c, a + b + c + d.

We can notice that the number of linear combinations by PBCs is more than the number of PBCs. This is an over-restrained problem. In addition, there is another over-restrained situation in this processing. The number of spectral lines converted from QBs to Fourier space is more than $(3^{n}-1)/2\;$. Therefore, if we want to find the proper solution of PBCs, other constraints from the physical model are needed to degrade the difficulty of calculation as follows: The $(3^{n}-1)/2\;$ beat components lines are picked out from MEM converted spectral lines. The maximum energy lines position of the beat component needs to be smaller than the sum of all the eigenenergy transition levels in the nuclear transition model of the sample. For ^57^Fe nucleus HFS, the maximum energy position of the spectral lines is smaller than the value of (a + b + c + d) in the calculation. With this constraint, the number of spectral lines is reduced from thousands to hundreds. Moreover, taking the intensity of these spectral lines into consideration, we only capture the mainly spectra lines stronger intensity than the average of the noise intensity and suppose that the beat components are among them. The noise is defined whose intensity is smaller than the 1% of the strongest intensity among all the spectral lines, which is common in EXAFS (Extended X-ray Absorption Fine Structure) analysis [27,28].The second restriction is to choose a reasonable energy region for the candidates of PBC. We assume that ‘M’ denotes the number of chosen spectral lines. ${}_{M}P_{n}$ is the permutation number of PBC combinations. If the candidates of PBC are not reduced, there is an exponential growing value of greatly increasing the calculation as the nucleus HFS becomes complex. Hence, supposing that the intensity of PBCs are higher than other spectral lines, the magnitude of the energy positions of these chosen spectral lines is used to restrain the region of the PBC. The PBC candidates which satisfy the constraints are less than hundreds.

After choosing the region of spectral lines, we obtain a large number of PBC lists. Each list is plugged back into Formula (1) and forms a column matrix of these calculation value. Every value in this matrix needs to be compared with all chosen spectral lines positions in order to find out the best list of PBCs which describe $(3^{n}-1)/2\;$ lines in the chosen spectral lines almost perfectly. This process is depicted by Equation (2): (2)$${\left(\begin{array}{cccc} 1 & 0 & \cdots & 0\\ 0 & 1 & \cdots & 0\\ 0 & 0 & \cdots & 0\\ 0 & 0 & \cdots & 1\\ \vdots & \vdots & \vdots & \vdots \\ 1 & 1 & \cdots & -1\\ \vdots & \vdots & \vdots & \vdots \\ 1 & 1 & 1 & 1 \end{array}\right)}_{\frac{3^{n}-1}{2}\times n}\times {\left(\begin{array}{c} a_{i}\\ b_{i}\\ c_{i}\\ \ldots \end{array}\right)}_{n\times 1}\times {\left(\begin{array}{ccc} 1 & \ldots & 1 \end{array}\right)}_{1\times M}-{\left(\begin{array}{ccc} E_{peak_{1}} & \cdots & E_{peak_{1}}\\ \vdots & \vdots & \vdots \\ E_{peak_{n}} & \cdots & E_{peak_{n}} \end{array}\right)}_{\frac{3^{n}-1}{2}\times M}\;={\left(\begin{array}{ccc} Res_{11} & \cdots & Res_{1n}\\ \vdots & \vdots & \vdots \\ Res_{nn} & \cdots & Res_{Mn} \end{array}\right)}_{\frac{3^{n}-1}{2}\times M}$$

The first matrix represents all possible linear combinations by PBCs, and the matrix size is $\frac{3^{n}-1}{2}\times n$. The second matrix is a list of selected PBCs where *i* means the No. *i* group from the candidates and the value *i* is $1\leq i\leq_{M}P_{n}$. ${}_{M}P_{n}$ is the possible permutations of PBC where M is the number of chosen spectral lines and *n* is the number of PBCs. The third matrix is a matrix that each element is 1 for reshaping the matrix. Then, the size of the minuend matrix in Equation (3) is $(3^{n}-1)/2\times$
*M.* In the fourth matrix, every line is the same, which is $E_{peak_{1}\;}$ to $E_{peak_{N}}$. $E_{peak_{1}}–E_{peak_{N}}$ is a set of the chosen spectral lines which contains the beat components. The left side of the equation means that every value of linear combinations generated by PBCs subtract every chosen spectral line. Then, we can get a residue matrix on the right side of the equation.

The residue matrix denotes the deviation between the linear combinations of the elements in the PBCs list and every chosen spectral line. We will choose the number of M minimum residues in different lines of this matrix. For easily understanding Equation (2), we take the nucleus HFS model with two prime beat components (PBCs) as an example to describe this process. It can be depicted by Equation (3). Here, we propose that there are six chosen spectral lines. It means that ‘M’ equals 6:(3)$$\begin{array}{l} \overset{{\left(\begin{array}{c} a\\ b\\ a-b\\ a+b \end{array}\right)}_{4\times 1}}{\overset{︷}{{\left(\begin{array}{cc} 1 & 0\\ 0 & 1\\ 1 & -1\\ 1 & 1 \end{array}\right)}_{4\times 2}\times {\left(\begin{array}{c} a\\ b \end{array}\right)}_{2\times 1}}}\times {\left(\begin{array}{cccccc} 1 & 1 & 1 & 1 & 1 & 1 \end{array}\right)}_{1\times 6}-{\left(\begin{array}{cccccc} E_{peak1} & E_{peak2} & E_{peak3} & E_{peak4} & E_{peak5} & E_{peak6}\\ E_{peak1} & E_{peak2} & E_{peak3} & E_{peak4} & E_{peak5} & E_{peak6}\\ E_{peak1} & E_{peak2} & E_{peak3} & E_{peak4} & E_{peak5} & E_{peak6}\\ E_{peak1} & E_{peak2} & E_{peak3} & E_{peak4} & E_{peak5} & E_{peak6} \end{array}\right)}_{4\times 6}\\ ={\left(\begin{array}{cccccc} a-E_{peak1} & a-E_{peak2} & a-E_{peak3} & a-E_{peak4} & a-E_{peak5} & a-E_{peak6}\\ b-E_{peak1} & b-E_{peak2} & b-E_{peak3} & b-E_{peak4} & b-E_{peak5} & b-E_{peak6}\\ \left(a-b\right)-E_{peak1} & \left(a-b\right)-E_{peak2} & \left(a-b\right)-E_{peak3} & \left(a-b\right)-E_{peak4} & \left(a-b\right)-E_{peak5} & \left(a-b\right)-E_{peak6}\\ \left(a+b\right)-E_{peak1} & \left(a+b\right)-E_{peak2} & \left(a+b\right)-E_{peak3} & \left(a+b\right)-E_{peak4} & \left(a+b\right)-E_{peak5} & \left(a+b\right)-E_{peak6} \end{array}\right)}_{4\times 6} \end{array}$$

In order to ensure picking out the minimum residue, we need to take the absolute value of the elements in the residue matrix and then search the global minimum in the matrix. After finding the global minimum element, this column and line will be excluded. The next global minimum value will be found in the next step. In addition, another column and line will not be reconsidered. This procedure repeats until we get M residue values in this matrix. Every group of PBC will obtain a column matrix of M minimum residue matrix. We can compare the M minimum residue’s mean and mean square error. Finally, the best list of PBCs with the minimum mean and standard variance of the residue column matrix is picked out.

The PBCs list obtained means the energy gaps of HFS. A total of $(3^{n}-1)/2\;$ spectral lines are generated by the combinations of the PBCs, which correspond to possible quantum beat components in the coherent NFS. In addition, the hyperfine structure of the nucleus can be inferred from the PBCs.

## 3. Case Study and Discussion

The temperature dependent NFS data were collected at KEK AR-NE3 beamline (Xiaowei Zhang, Tsukuba, Ibaraki, Japan), and the experimental detail—see references [13,29]. The AR(Advanced Ring) machine is a single bunch ring, and the bunch interval is 1.2 μs. It is a suitable ring for ^57^Fe NFS experiments. Our sample is ^57^Fe_2_O_3_ powder with natural iron. The content of Fe is about 1 mg/cm^2^. Therefore, we make the sample as proper thickness, and the thickness effect can be ignored. As a case study, we analyzed the time-domain NFS spectra of α-Fe_2_O_3_ with the PBC extraction method after MEM conversion at 200, 230, 260, 290, 298, and 310 K. The list of PBCs was picked out from a bunch of spectral lines. According to the ^57^Fe nucleus HFS, there are four PBCs (a, b, c, d, as shown in Figure 1). After a calculation, the 40 beat components generated by the PBCs correspond to the major spectral lines in the energy spectra. The results of the prime spectra lines are tableted at different temperatures are given by the PBC extraction method. By converting the PBCs into the differences between ^57^Fe nucleus energy levels, we notice that the differences change with temperature. Moreover, we observe that the energy gaps of b and d (as Figure 1a shown) turn over below and above Morin temperature. This result further proves the effectiveness and reliability of the PBCs method. 

### 3.1. Analyzing the Time-Domain NFS Spectra of Hematite at the Room Temperature

Firstly, we analyzed the time-domain NFS spectra of hematite by the PBC extraction method at room temperature. The NFS spectrum observed is shown in Figure 3a. The logarithm of NFS spectrum and fitting baseline are shown in Figure 3b. After eliminating the baseline, the fringes of quantum beats that carry the information of interference between the energy levels of ^57^Fe HFS are extracted as shown in Figure 3c. The next step is to convert the time spectrum to frequency space and extract the beat components. The conversion results of energy spectra are shown in Figure 4. In the energy space, the spectral lines are the beat components which correspond to the interference among the energy levels of the hyperfine structure.

After getting the beats’ frequency lines, we need to pick out the PBCs. In a powder hematite sample, the transition model of ^57^Fe is shown in Figure 1a, and there are four energy gaps in its HFS. This means that the number of PBC is four. Because of the maximum spectral line position is around 0.8 μeV [30], we choose the region of interest (ROI) from 0 to 1 μeV, which includes all the spectral lines with strong intensity (Figure 4), and assume that the all beat frequencies will be generated by the liner combinations of PBCs. In Figure 4, the intense peak corresponds to a strong interference between two energy levels. Thus, we choose an intensity threshold of the beats’ components that is equal to the noise intensity which is around 1.6 × 10^−4^, and select four PBCs from the 0.1 to 0.32 μeV range that has spectral lines with higher intensity.

In this case, the permutation number of a possible list of PBCs is ${}_{70}P_{4}$ which is about 2.2 × 10^8^. In order to reduce the calculation amount, we suppose that the intensity of PBCs has higher possibility appeared in the region of spectral lines with higher intensity. Therefore, the region from 0.1 to 0.32 μeV that has spectral lines with higher intensity is chosen. The permutation number of the list of PBCs in this region is reduced to hundreds of choices. With the PBC extraction method described in Section 2.2, four PBCs are extracted. They are 2.96 × 10^−7^, 1.79 × 10^−7^, 1.69 × 10^−7^ and 1.57 × 10^−7^ eV, and correspond to a, b, c, d in HFS of ^57^Fe, as shown in Figure 1a. The 40 beat components are assigned by the four PBCs as shown in Figure 5. The blue lines represent the spectral lines from MEM conversion of the original experimental spectrum, and the red lines represent the energy positions of the beat components generated by the PBCs. The combination labels represent the spectral lines that we have assigned. In Table 1, the specific information of Figure 5 is presented. The average value and standard deviation of the deviations between PBCs generated components, and corresponding beat components are 0.0016% and 6.6%, respectively.

According to the results of Figure 5 and Table 1, we observe that 95% of intense beat components in the spectrum are explained by the combinations of the four prime beat components perfectly. Some of the combinations, such as the component (a − b + d) and (a + c − d), do not find the adaptable spectral lines with small deviations. We speculate that these are probably due to not completing nuclear HFS, and we should use the F quantum number in the nuclear hyperfine structure like the atomic case. For α-^57^Fe_2_O_3_, we need to consider the coupling between the ^57^Fe^+3^ atomic angular momentum J and the nuclear spin I. This quantum number should be the good quantum number in the complete HFS of ^57^Fe^+3^. In addition, other reasons may come from dynamical beats that we probably do not consider, the background that we removed, statistical error and other experimental reasons.

### 3.2. Analyzing NFS Spectra of Hematite below/above the Morin Temperature by the PBC Extraction Method

After analyzing the NFS spectra of hematite at room temperature, we continue to resolve the NFS spectra at different temperatures of 200, 230, 260, 290 and 310 K. Hematite is a magnetic material, and there are two different antiferromagnetic states around its Morin point (260 K): antiferromagnetic below 260 K and weak ferromagnetic above 260 K [31]. This phenomenon is shown on the MS that the values of the energy transition levels’ intervals b, d of the excited state turn over. This result has been observed from the conventional Mössbauer spectroscopy results in the temperature region [32], as the hollow squared point in Figure 6 shown.

We apply the PBC extraction method to the NFS spectra of the hematite sample at different temperatures and observe the reversal of b and d across the Morin transition [33], as the solid dots in Figure 6 shown. The same conclusion is reached by comparing these two results as Figure 6 shown. It means that the PBC extraction method is able to derive the correct results from QB spectra of hematite when its magnetic property changes during the Morin transition.

In order to compare with conventional Mössbauer spectroscopy result, the PBCs are transformed to the traditional six absorption transition lines as shown in Figure 1. Lines 1–6 is represented by PBCs as 0, b, b + c, a + b, a + b + c, a + c + b + d. Equation (4) is used to transform the energy to velocity (mm/s):(4)$$∆E=E_{\gamma }vc^{-1}$$
where ∆E is the energy difference between spectral lines; $E_{\gamma }$ is the incident energy (e.g., for hematite, $E_{\gamma }=14.4\;\mathrm{keV}$); *c* is the velocity of light. The transformation results are shown in Table 2 and the transition line positions obtained are in a reasonable range.

## 4. Conclusions

In summary, we propose a PBC extraction method to analyze the energy spectrum after converting the time-domain NFS spectrum by the idea of solving the inverse problem. With this method, the PBCs are extracted from chosen spectral lines and the $(3^{n}-1)/2\;$ beat components can correspond to the spectral lines. These components match the majority of chosen spectral lines well in energy space and can assign the physical meaning of the spectral lines with high intensity, which are the interference character of the quantum beats in time-domain NFS spectra.

We take the NFS spectra of powder hematite sample at several temperatures as a study case. At first, we extract the four PBCs and describe 40 beat components that are combined by four PBCs in the energy spectral lines at room temperature (295 K). Then, the NFS spectra of hematite above and below Morin temperature are analyzed by the PBC extraction method. The calculation results of the PBCs at different temperatures are consistent with the magnetic phenomenon of hematite. It proves that this method is effective to get the hyperfine structure of the sample. In addition, the PBCs extracted could be converted to the conventional MS style with six spectral lines and analyzed as the conventional MS result. In this case study, the PBC extraction method has worked well. This success denotes that we can deal with the NFS spectra analysis by this method.

This is the first time to explain the meaning of 40 beat components of hematite which is converted from NFS spectrum. However, there are still some lines we could not explain, which probably can be attributed to the hyperfine structure that does not consider the total quantum number F or other reasons which come from the data processing, experimental reasons and so on. There are two limitations to this method. One is that the “initial guess” on the number of PBCs for unknown HFS of the sample. However, this limitation can be ignored for known HFS of the sample. The other one is that, as the HFS becomes more complicated, the number of PBCs (energy gaps of HFS) increases. Then, the linear combinations of PBCs are exponentially increasing, and the computation consumption will be overwhelming since there will be $(3^{n}-1)/2\;$ linear combinations of PBCs. Therefore, we are limited by the computation power at present. Some of the spectral lines can be explained by the PBC extraction method. Some of the spectral lines may be missing. Even though we only focus on the spectral lines with stronger intensity, all the linear combinations of PBCs need to be ergodic until finding the best fit for all. In the future, we will optimize our algorithm and decrease the complexity of PBCs to prove the method feasibility. For the next step, we will focus on refining the hyperfine structure by taking the quantum number F into the model and hope to explain more spectra lines in energy space. In addition, the line intensity ratio and thickness effects are of significant interests and will be addressed in the future improvement of our approach.

## Figures and Tables

**Figure 1 materials-12-01657-f001:**
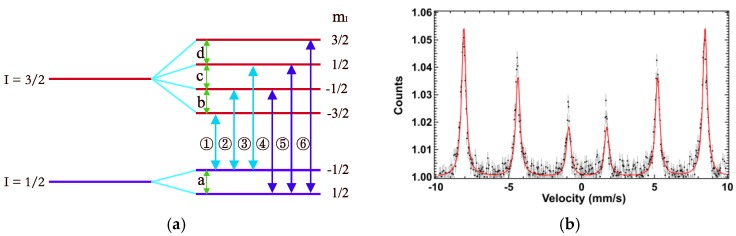
(**a**) HFS of ^57^Fe nucleus (a, b, c, d are called ‘prime beat components’, they represent the energy gaps between energy levels of HFS; Line ①~⑥ represent the six allowed transitions; m_I_ represents the nucleus magnetic quantum number after Zeeman-splitting); (**b**) the conventional MS of Fe_2_O_3_ (the black dots represent the experimental data of conventional MS and the red multi-peak curves are the fitting result of experimental data) [24].

**Figure 2 materials-12-01657-f002:**
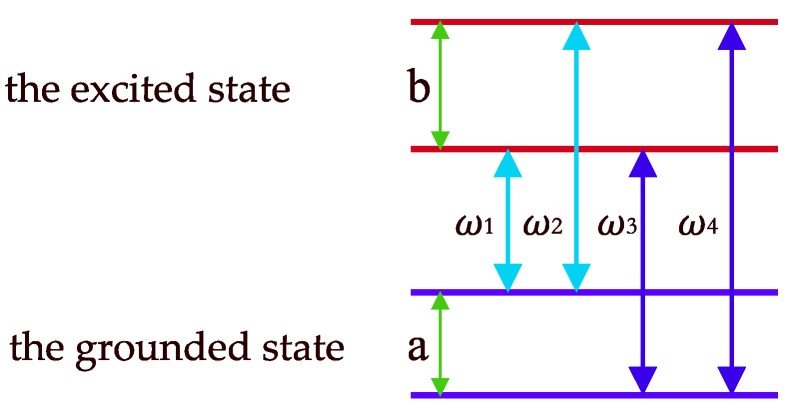
An example of nucleus HFS model with two prime beat components (PBC).

**Figure 3 materials-12-01657-f003:**
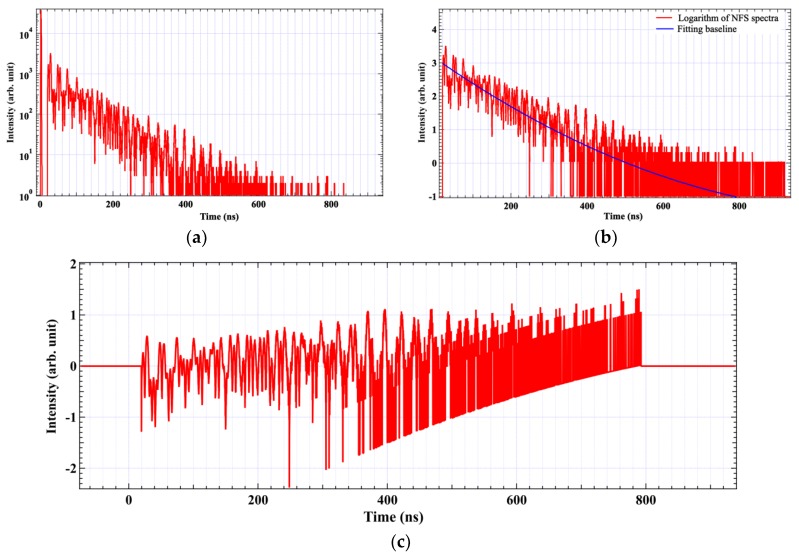
(**a**) the NFS spectrum of hematite at room temperature; (**b**) the red line represents the logarithm of NFS spectra and the blue line represents the fitting baseline by polynomial function; (**c**) the fringes of quantum beats after isolating the baseline.

**Figure 4 materials-12-01657-f004:**
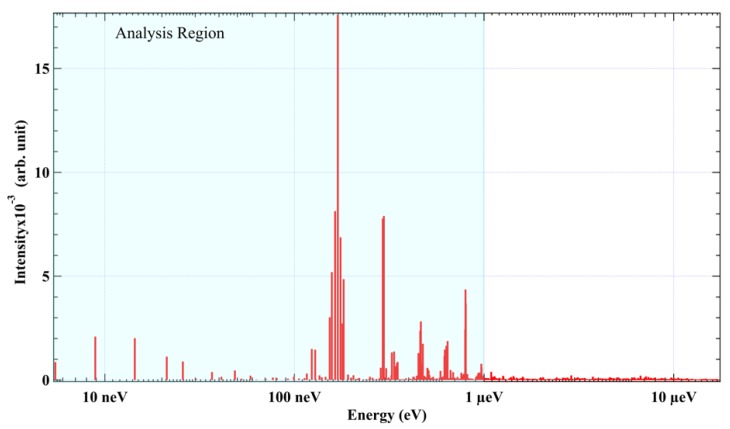
Spectral lines of hematite after MEM conversion.

**Figure 5 materials-12-01657-f005:**
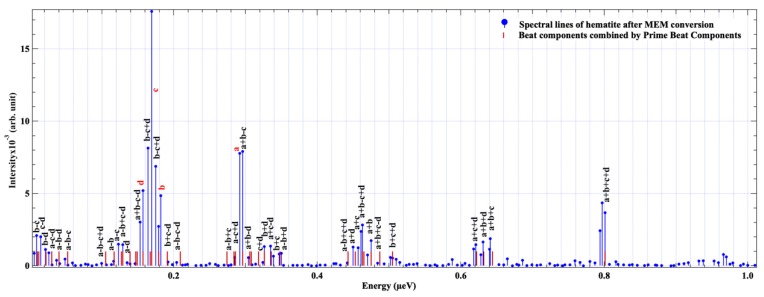
The PBCs and 40 beat components of ^57^Fe_2_O_3_ powder at room temperature acquired by PBCs extraction method (the blue lines mean the spectral lines after MEM conversion; the red lines mean the beat components that are combined by PBCs; the combinations are labeled above the spectral lines assigned).

**Figure 6 materials-12-01657-f006:**
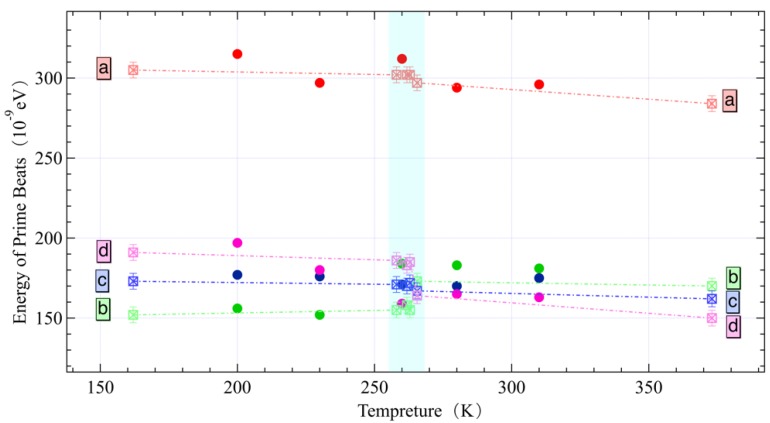
Comparison between the PBCs calculated by the PBC extraction method (represented by the solid dots) and the conventional Mössbauer spectroscopy measure result (represented by the hollow squared point); the blue region represents the Morin transition region at 260 K.

**Table 1 materials-12-01657-t001:** Forty beat components generated by PBCs and contrast with the spectral lines from the QB spectrum.

List of QB Components Generated by PBCs		Beat Components (eV)		Deviations (eV)
Index	Combinations	Generated by PBCs	Original Spectral Line	
1	b − c	1.00 × 10^−8^	8.89 × 10^−9^	1.11 × 10^−9^
2	c − d	1.20 × 10^−8^	1.44 × 10^−8^	2.40 × 10^−9^
3	b − d	2.20 × 10^−8^	2.12 × 10^−8^	8.00 × 10^-10^
4	a − c− d	3.00 × 10^−8^	2.58 × 10^−8^	4.20 × 10^−9^
5	a − b− d	4.00 × 10^−8^	3.67 × 10^−8^	3.30 × 10^−9^
6	a − b− c	5.20 × 10^−8^	4.85 × 10^−8^	3.50 × 10^−9^
7	a − b− c + d	1.05 × 10^−7^	9.93 × 10^−8^	5.70 × 10^−9^
8	a − b	1.17 × 10^−7^	1.16 × 10^−7^	1.00 × 10^−9^
9	a − c	1.27 × 10^−7^	1.23 × 10^−7^	4.00 × 10^−9^
10	a − b + c − d	1.29 × 10^−7^	1.29 × 10^−7^	0.00
11	a − d	1.39 × 10^−7^	1.35 × 10^−7^	4.00 × 10^−9^
12	a + b − c − d	1.49 × 10^−7^	1.53 × 10^−7^	4.00 × 10^−9^
13	d	1.57 × 10^−7^	1.57 × 10^−7^	0.00
14	b − c + d	1.67 × 10^−7^	1.64 × 10^−7^	3.00 × 10^−9^
15	c	1.69 × 10^−7^	1.69 × 10^−7^	0.00
16	b − c − d	1.47 × 10^−7^	1.75 × 10^−7^	2.80 × 10^−8^
17	b	1.79 × 10^−7^	1.79 × 10^−7^	0.00
18	b + c − d	1.91 × 10^−7^	1.92 × 10^−7^	1.00 × 10^−9^
19	a − b − c − d	2.09 × 10^−7^	2.04 × 10^−7^	5.00 × 10^−9^
20	a − b + c	2.86 × 10^−7^	2.85 × 10^−7^	1.00 × 10^−9^
21	a −c + d	2.84 × 10^−7^	2.92 × 10^−7^	8.00 × 10^−9^
22	a	2.96 × 10^−7^	2.96 × 10^−7^	0.00
23	a + b − c	3.06 × 10^−7^	3.04 × 10^−7^	2.00 × 10^−9^
24	a + b − d	3.18 × 10^−7^	3.24 × 10^−7^	6.00 × 10^−9^
25	c + d	3.26 × 10^−7^	3.26 × 10^−7^	0.00
26	b + d	3.36 × 10^−7^	3.35 × 10^−7^	1.00 × 10^−9^
27	a + c − d	3.08 × 10^−7^	3.39 × 10^−7^	3.10 × 10^−8^
28	b + c	3.48 × 10^−7^	3.47 × 10^−7^	1.00 × 10^−9^
29	a − b + d	2.74 × 10^−7^	3.50 × 10^−7^	7.60 × 10^−8^
30	a − b + c + d	4.43 × 10^−7^	4.42 × 10^−7^	1.00 × 10^−9^
31	a + d	4.53 × 10^−7^	4.50 × 10^−7^	3.00 × 10^−9^
32	a + c	4.65 × 10^−7^	4.61 × 10^−7^	4.00 × 10^−9^
33	a + b − c + d	4.63 × 10^−7^	4.63 × 10^−7^	0.00
34	a + b	4.75 × 10^−7^	4.75 × 10^−7^	0.00
35	a + b + c − d	4.87 × 10^−7^	4.84 × 10^−7^	3.00 × 10^−9^
36	b + c + d	5.05 × 10^−7^	5.05 × 10^−7^	0.00
37	a + c + d	6.22 × 10^−7^	6.21 × 10^−7^	1.00 × 10^−9^
38	a + b + d	6.32 × 10^−7^	6.31 × 10^−7^	1.00 × 10^−9^
39	a + b + c	6.44 × 10^−7^	6.41 × 10^−7^	3.00 × 10^−9^
40	a + b + c + d	8.01 × 10^−7^	8.01 × 10^−7^	0.00

**Table 2 materials-12-01657-t002:** The energy transition lines converted by PBCs at different temperatures.

	Temperature Energy	200 K	230 K	260 K	280 K	310 K
Line (mm/s)						
Line 1		0	0	0	0	0
Line 2		3.25	3.17	3.84	3.82	3.77
Line 3		6.95	6.85	7.41	7.37	7.43
Line 4		9.83	9.37	10.35	9.96	9.96
Line 5		13.53	13.04	13.92	13.95	13.61
Line 6		17.64	16.81	17.24	16.95	17.01

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
