# Peer review of "The Prime Beat Components Extraction Method for the Time Spectra Analysis of Nuclear Resonant Forward Scattering"

_materials, 2019, doi:10.3390/ma12101657_

Round 1

Reviewer 1 Report

The manuscript titled 'The prime beat components extraction method for the time spectra analysis of Nuclear Forward Scattering' describe new approach for evaluation of of Nuclear Forward Scattering (NFS) of synchrotron radiation. The manuscript describe the transformation of  NFS time spectra from time domain in which the experimental data is acquired to energy domain. This transformation based on the analysis  of main quantum beats is proceed by the a Maximal Entropy Method (MEM).

Next, the energy spectrum is analyzed by a combining of all possible pairs of interfering photons.

The topic of the article is of high importance, because NFS is powerful method for investigating  of not  only iron containing material. However, the evaluation process of NFS time spectra is challenging task mainly when more phases are present in the investigated material. Therefore, I appreciate much the effort of the authors focused on NFS data analysis methods.

In my opinion the transformation of primary beats to energy  domain is a revelatory idea, however before the publication in 'Materials' several parts, statements should be clarified and improved.

The comments can be divided into two groups, principle questions/comments and minor erros and mistakes.

Questions/comments:

1) As I am not a native speaker I am not the best one to comment the language. However, in my opinion several misunderstandings can origin from some language inaccuracies.

For example: I don't understand the sentence on line 49. How it is connected with the text ?

2) In the introduction I miss a remark  or comment on the direct transformation from time to energy domain connected with the Phase problem in NFS stated for example by W. Sturhahn, PRB 63, 094105.

3) Some more clear definition of the term “prime beat  components” could be useful.

4) In the Figure 1 there is an upper part which is not explained neither in the text nor in the figure caption. What does this part mean ?

5) line 79: The expression nucleus transition model is not commonly used. Could it be replaced by some other term ?

6) lines 92 – 93 the sentence is not very clear.

Could be the sentence ”It means that if the interference spectra are analyzed independently, we ought to subtract the fitting curve to eliminate the influence of the incident beam by logarithm.” reformulated ?

7) I understand that the “exponential” is subtracted from the time spectra to get solely the beatings. It is not clear if this operation is based on some theoretical description ? The quantum beats in the time spectrum occurs due to the an interference of photons, which are theoretically described by complex field intensity. In order to get the scattered field the sum of complex amplitudes should be performed and finally the detected intensity is a square of this complex sum. In some special cases the resulting amplitude can be described by a sinusoidally modulated exponential function. In such case the exponential can be subtracted to get the oscillations. I doubt that this is valid for any case. Could the authors comment this ?

8) The quantum beats, especially in case of samples exhibiting large magnetic splitting, look not like an exponential function with some oscillation. Often, they drops near to zero.  In the manuscript there is no description how to find the subtracted function. Probably some figure of the time spectrum together with the removed function can be useful. Good solution can be to add the removed exponential function to the Figure 2(a).

9) lines 120-125: The description of number of combination is not very clear and the equation (2) is not equation. It is difficult to understand what does this notation mean.

10) line153 : What does the equation mean ? There is a summation of the residua ? What is the third term ? That is matrix with 1 in each position ?

It should be clarified.

11) The FFT in case of NFS cannot be performed directly because the information on the phase is missing. It can be done only from amplitudes like is shown in [20]. Usually, there is not enough point  in the time pattern to get sufficient energy resolution in energy domain. The authors applied MEM method for the transformation to energy domain. Can the author comment the advantages of these approach in more detail.

12) line 193:  In energy spectrum, high intensity is related to high probability of

interference of energy levels. Thus, 70 spectral lines with higher intensity in this energy region are

chosen to be the alternative options for the PBC.

Thus it means that 70 highest intensity in the spectrum is only considered for further evaluation ? How was the threshold  found ?

13) lines 180-212: The procedure of “labeling” of PBC to a spectral amplitudes is not clearly described.

Could be the labeling preformed also in case of two sextets in the spectrum ? Or in any combination of more phases in the sample ?

14) After labeling of energy spectrum the authors performed fitting of PBC to get the level splitting. What are the errors of the determined PBC or line positions ?

15) The authors stated that the method can convert the time spectrum to energy domain which can be evaluated in the same manner like Mossbauer absorption spectrum. They refer to Figure 1. The spectrum shown in the figure 1 is really that one obtained from the transformation procedure ? Or the red line in Figure 1 is a calculated spectrum for the nuclear level splitting according to PBC ? And what about the amplitudes of individual lines ? They are considered to be 3:2:1:1:2:3 ? Is the possibility of  3:x:1:1:x:3 scheme taken into a consideration ?

16) In the manuscript the line positions are considered for the evaluating of the PBC. But the authors did not discussed the intensities of the time spectra transformed to energy domain.

16) line 159 – 261: “It is the first time to explain the physical meaning of the spectral lines converted from NFS spectrum to energy domain. However, there are some converted spectra that we could not explain, which probably attributes to the incomplete hyperfine structure theory.”

It is not clear what does the “ incomplete hyperfine structure theory” means ?

17) In conclusion the authors stated that this method can be applied to convert the NFS signal from the time to energy domain where it can be farther analyzed. Nevertheless, this is not demonstrated in the manuscript. The application of this method is demonstrated on one example of hematite which exhibits magnetic splitting resulting in 6 Mossbauer lines. That is relatively simple spectrum which can be easily analyzed in time domain using CONUSS, MOTIF, EFFINO etc. The important question which is not discussed in the manuscript is whether this method is suitable for analyzing of the samples with more spectral components (two sextets, two sextets, doublets etc. ). My suggestion is to try such procedure also on such type of time spectra. Which, moreover, can be generated “theoretically” just as a model for testing the procedure.

Minor corrections:

before ( and [ brackets and behind a dot on the end of sentence should be a space. That occurs in the manuscript many times....

line 27: smaple / sample ?

Line 30: element vs.  isotope ?

Line 43: I am not sure whether the references 14-17 really describe the transformation from energy to time domain.

In conclusion: I my opinion the manuscript contain several interesting ideas which should be further evolved. However, they are not presented in a clear manner. Therefor, before publication, I would suggest to improve ambiguous parts.   

Author Response

Thanks for your comments and help. We revised our manuscript according to your suggestion. Please check the reply in the attachment. 

Reviewer 2 Report

The nuclear forward scattering becomes important in the field of Moessbauer spectrocopy.  But the comprehension is not easy.  Li and Zhang proposed a new method to understand quantum beats of the nuclear forward scattering.  The method was confirmed experimentally by checking the Morin transition of hematite.  This new method is meaningful and worth publishing.

Author Response

Thanks for your comments and help. We revised our manuscript according to your suggestion. Please check the new version of this manuscript. 

Reviewer 3 Report

Referee report

The manuscript reports new method for the analysis of the time spectrum of nuclear resonant forward scattering. The authors proposed that the energy level of the hyperfine structure can be directly determined by the quantum beat of the time spectrum by the proposed method called as the prime beat components method.

   The methodology is quite new and interesting. However, since the manuscript is poorly written in the points of English and flow of the story, it is difficult to follow the arguments.

In addition, I have some comments and questions, which relates to the fundamental and essential points of the manuscript.

First, please answer the following my comments. After modification of manuscript, I strongly recommend authors to carefully read the manuscript and prepare manuscript with less careless misses. “Please prepare revised manuscript much more carefully.” Then, please ask native English speaker for the correction of English.

Examples of careless misses

(Page 1, line 27) “smaple” instead of “sample”

(Page 2, Fig.1a) Authors didn’t explain the meaning of Δm, mI and I.

(Page 3, line 88) “pulse12” instead of “pulse[12]” ?

(Page 3, line 102) Authors didn’t explain tau_0.

(Page 4, line 127) Only 5 transitions are listed among 6 transition. “0” should be added?

Examples of poor explanation

(Page 3, line 100) Please explain the MEM method at least the summary of the method. Otherwise readers cannot understand why the method is more convenient and accurate than FFT.

(Page 3, line 121) Please explain how to obtain (3^n-1)/2. Otherwise, we don’t know if the formula is valid only for the case of 57-Fe or valid for all nuclear resonant species. How did you consider the selection rule of the transition?

(Page 6, line 212) Please explain about the quantum number F. Do you mean the quantum number representing the coupling of the net orbital angular momentum and the net nuclear spin?

More essential and critical comments.

(Page 3, line 93) Authors wrote “In order to simplify the calculation, we use the polynomials instead of the exponential function to fit the baseline of quantum beats.”. I don’t understand why exponential function is not used because the calculation is still simple in the case of exponential function. Please explain more. In addition, please show the baseline in figure 2(a).

(Page 3, line 95) Authors wrote “Then the residual between the original quantum beats and the fitting curve, which contains the interference fringes basically, is withheld as shown in the Figure 2(b).”. However, it seems not the data shown in figure 2(b) is obtained by just subtraction of the baseline from the time spectrum. Please show the exact treatment performed to obtain the date shown in figure 2(b).

(Comment on methodology) From the Mossbauer spectrum shown in figure 1(b), the effective thickness of the Fe2O3 sample is not so high. Therefore, I expect that the exponential function can be assumed for the baseline of the time spectrum. However, in thicker sample, the baseline departs from the exponential; instead, dynamical beat must be considered. In such case, your procedure of baseline subtraction to obtain the fringes of the quantum beat doesn’t work. This is because each Mossbauer line shows different thickness. For example, Mossbauer lines 1 and 6 is more intense and the corresponding speeding up effect is stronger causing more prominent dynamical beat comparing with other Mossbauer lines. Therefore, in general cases, the time spectrum is very complicated as showing the hybrid beat of quantum and dynamical beats and the baseline cannot be defined in general sample. In my opinion, the method you proposed is valid only for sample in the thin limit.

(Comment on methodology) Another comment on the methodology is that authors didn’t consider the polarization. To accurately calculate the time spectrum in general cases, the polarization of the gamma rays must be considered. Please provide a way to consider it. 

(Comment on methodology) In figure 5, the obtained PBS is shown. Usually, accuracy of the time spectrum is higher than that of the Mossbauer spectrum. Therefore, the hyperfine parameters obtained by the time spectrum should be more accurate than these by Mossbauer spectrum. However, in the figure, the points distribute largely comparing with the standard deviation of the points by the Mossbauer spectrum. This result suggests that the proposed analysis method loses the information of the time spectrum and not effective method comparing with previous analysis methods for the time spectrum.

(Comment on methodology) Please provide standard deviation of the points in figure 5 and show how to evaluate it. Showing the static standard deviation is mandatory for any methodology.

(Comment on methodology) If you cannot answer the above essential comments, it means that your method is unfortunately not generally useful one. However, I believe that your new approach is still valuable as the first step. Please provide the limitations of your methodology and possible ways to overcome the limitations in the future.

Author Response

Thanks for your comments and help. We have revised our manuscript according to your comments. Please check the reply in the attachment.

Reviewer 4 Report

This article describes an ambitious analysis method for the time spectra of nuclear resonant forward scattering (NFS) of synchrotron radiation (SR), which is widely measured in SR facility, such as American Photon Source (APS) in USA, European Synchrotron Radiation Facility (ESRF) in EU, and Super Photon ring 8 (SPring-8) in Japan. In experimental viewpoint, the time spectra of NFS is an excellent method for the evaluation of hyperfine structure of nuclei, through which we can detect the valence, magnetic order, and other element (actually nuclear)-specific information of atoms. Especially this nuclear-specific property is unique method because it might be applied as isotope probes in condensed matter physics. However, the analysis of the time spectra is somewhat complicated and not intuitive, due to mainly the loss and excess of some information and partly its appearance. Meanwhile, as described in this article, the recent great development of computational method in information science has possibility to solve this problem on time spectra of NFS. The authors now trying to alleviate this problem in this article by making maximum use of experimental data and physical conditions, as well as the computational method including Maximum Entropy Method (MEM). Consequently, the aim of this article is highly important and this result is a good milestone to be published. However, some description is unclear to me and considering that MDPI material does not limit the article length, I recommend publication after a major revision. Following is my comments on each subject.

1) History to this research

1-1) In line 23-32, authors describe the energy-domain Mössbauer spectra with both radioactive isotopes and SR. As Prof. Zhang well knows, synchrotron Mössbauer source, developed by Prof. Gerdau using nuclear Bragg monochromator by Prof. Smirnov is a critical tool and should be referred.  

1-2) Please check again the references. For example, Refs. 4 and 5 are articles using time spectra and not energy spectra, although this reference are shown in the description on energy-domain method. In addition, the reference on multi-nuclide SR Mössbauer spectroscopy used in ref. 3 is originally developed by Prof. Seto in 2009 (PRL, vol. 102, No.217602) and the PRL article should be referred rather than ref. 3. If authors would like to refer the K-40 Mossbauer energy spectrum, Nakano 2015 (PRB, Vol. 91, No. 140101R) might be also useful. This is only an example and please check the other references, such as references in line 43.

2) Method of analysis

2-1) The main problem in the analysis of the NFS time spectra is two, in my opinion. One and essential problem is the lack of phase information of scattered photon field in measured time spectra. Another problem is the mixing of quantum beats and dynamical beats, which is caused by the effective thickness. If these background conditions are shown in subsection 2.1, it is kind to readers.

2-2) On the background subtraction, written from line 87 to 96, the concept is very good, but you should also consider the dynamical beat, which is another reason for beat patterns in NFS time spectra. As you assumed in line 92, the quantum beat is dominant in time spectra and approximation for the time dependency written in the beginning of line 92 is right, only if the time range is sufficiently below its lifetime (t << tau0) or if the effective thickness, which causes the dynamical beat, is very low. The background subtraction is also one of major issues in XAFS analysis, because this process make ‘fake’ oscillation signals, which sometimes meets the desire of researchers.

2-3) Section 2. 3 is the main subject of this article, considering ref. 30. The concept is excellent; the MEM is the most ‘likely’ interpolating method if the obtained spectra is well-defined in all complex plane. Therefore, the MEM spectra often shows many ‘fake’ peaks which correspond to the linear combination of true energy values. The authors distinguished the true energy values by the method in section 2.3. However, its description is not kind to readers, especially around the explanation of Eq. (3). Is there any relation between n and N? What is the third matrix in Eq. (3)? (The authors jump from the explanation of second matrix to fourth matrix.) What the index ‘i’ means? (I think it might be labels for other groups of PBC, which means other nuclear transition model.) Why authors do not write A^n_M instead of _MP_n. the standard permutation symbol, if it shows standard permutation?

3) Condition of the experiment

3-1) Please show the thickness of sample. This directly relates the effective thickness, which causes the dynamical beats, another origin of the beat pattern in NFS time spectra.

4) Discussion part

4-1) On the discussion in lines 210-212, authors discuss the extra spectral lines in MEM spectrum. I suggest that authors should also think the possibility that this is the ‘fake’ components, caused by dynamical beats, polynomial background subtraction, statistical error, and other experimental and methodological reasons. This might affect the description in lines 259-260.

4-2) I also suggest the authors that they add the discussion of the limitation of this method and further clues for improvement. This method is applicable to observe the change in somewhat known samples, such as phase transition in hematite, but seems to be still hard to be applied to the analysis of unknown complex samples including two or more component. If you have apparent direction to improve this problem, it is very kind to readers to show the direction, if it is not the critical and unique idea for the next manuscript.

5) Minor points

Line 68 and others: “nucleus transition” should be “nuclear transition”.

Fig. 1 (b): It seems that the data in upper side of this graph shows too low value in the range except around the six peaks, if it shows the residual between the experimental data and fitting line. Please explain the meaning of this data.  

Line 173-174: Please appeal the advantage of the KEK AR-NE3 beamline. The long time period over 1 micro seconds is advantageous for Fe-57 NFS experiments, whose lifetime is 141 ns.

Author Response

(The authors gave the same response as above.)

Round 2

Reviewer 1 Report

I would like to thank the authors for considering all my comments and remarks. I think that more less all unclear points were clarify. Nevertheless, in the point 4. The question was focused to the Figure 1b. Please, remove or explain the upper part of the figure. That is the comparison with conventional MS spectrum ? Referenced at line 290 ? If yes, I thing it should by explained in the text.

    I am a little bit skeptic in the application of this method for evaluation of time spectra of unknown hyperfine strucuture. Nevertheless, I think that is very important first step toward the new evaluation approach of NFS.

Therefore, after polishing minor typographical mistakes, I recommend the manuscript for publication.

Author Response

Thanks for pointing out. 

This is not the comparison between the conventional MS and time-domain spectra in our approach. To avoid misunderstanding, we deleted the upper part of Figure.1b which is the residual between the fitted and measured conventional MS spectra. And we clarify it more clearly on line 105 -106. 

Reviewer 4 Report

Owing to the authors’ effort, their manuscript are quite improved. However, it has still some descriptions to be improved.

1) Method of analysis

1-1) Fig. 1 (b): Authors describe still nothing about the data in the upper side of this graph, that is, the dots shown with the scale [-5 x 10^3, 5 x 10^3]. They seems to be the residual, but they looks too small in the intermediate region between the peaks in the CEMS spectrum in the lower side.  

1-2) Is the 1 % standards for the discrimination of noise in line 170 is general or common one (maybe in XAFS analysis)? I suggest that you refer the reason of this standard.

1-3) The explanation on equation 3 is quite improved. Now I can understand the steps. Nonetheless, it seems insufficient or includes some mistakes. I am afraid of the following points. For the third matrix, including only 1, its dimension is 1 x M. For the Fourth matrix, including Epeak components, its dimension is (3^n-1)/2 x M and the component in the last column is Epeak_M. For the fifth matrix, showing the residual components Res, is also (3^n-1)/2 x M.

2) Condition of the experiment

2-1) In line 208, the reference 13 seems to have no relation to the experimental details at KEK. I am afraid you intend to refer another reference.

2-2) In my opinion, the thickness of 1 mg/cm^2 is not thin for enriched hematite. It corresponds to the effective thickness of 8, which shows non-negligible thickness effect. If the sample was natural hematite, I agree that the thickness effect is negligible. Please check the enrichment of the sample.

3) Discussion part

3-1) This method described in this manuscript is important milestone in the fusion of NFS and computational science. However, some limitations still remain in this method in the present stage in principle, except the limitation by PC power. It is partly described, but not sufficient. The necessity of appropriate “initial guess” on the number of PBC should be referred, although it might be solved in future.

4) Minor points

Line 71: “in measure time spectra” should be “in measured time spectra”.

Line 94: “are defined the shortest list” should be “are defined as the shortest list”.

Line 96: “nucleus hyperfine structure” should be “hyperfine structure of nucleus”.

Line 113: “principal” should be “principle”.

Line 146: “a = (omega1 – omega2)” should be “a = (omega2 – omega3)”.

Line 163: “MEM conversion spectral lines” should be “MEM converted spectral lines”

Line 320: I think “3^n 1” should be “3^n-1”. For some reason, the minus sign is vanished in the pdf version of your manuscript.

Line 327: “efforts” should be “effects”

Line 406: “${\mathrm{Fe}}^{57m}$” should be corrected.

I strongly suggest that you check all of your manuscript before your submission.

5) Comments for future (no relation to my judge in this review)

I would like to add one comments, although it does not relates to my judge in this review because I did not pointed it out in the previous review comments. It is on the error estimation in this method. I suggest authors considering the way to estimate the errors of the values obtained by this method. Phenomenologically, it explains the dispersion of the values in Figure 6 except the Morin temperature.

Author Response

Thanks for your valuable comments to improve the quality of this work. According to your suggestions, we revised this manuscript again. The response can be checked in the attachment.
